



# Validation analysis of deriving acetonitrile ($CH_3CN$) profiles by observations of SMILES from the International Space Station, in the stratosphere and lower mesosphere

Tamaki Fujinawa[1], Tomohiro O. Sato[1], Takayoshi Yamada[1], Seidai Nara[1,2], Yuki Uchiyama[1,3], Kodai Takahashi[1,3], Naohiro Yoshida[4], and Yasuko Kasai[1]

[1]National Institute of Information and Communications Technology, 4-2-1 Nukui-kitamachi, Koganei, Tokyo 184-8795, Japan
[2]University of Tsukuba, 1-1-1 Tennnoudai, Tsukuba, Ibaraki 305-8577, Japan
[3]Tokyo Gakugei University, 4-1-1 Nukui-kitamachi, Koganei, Tokyo 184-8501, Japan
[4]Earth-Life Science Institute, Tokyo Institute of Technology, IE-1, 2-12-1, Oookayama, Meguro-ku, Tokyo 152-8550, Japan

**Correspondence:** Y. Kasai (ykasai@nict.go.jp)

**Abstract.** Acetonitrile ($CH_3CN$) is one of the volatile organic compounds (VOC) and a potential tracer of biomass burning. We evaluated the capability of using observations derived from the Superconducting Submillimeter-Wave Limb-Emission Sounder (SMILES) on the International Space Station (ISS) to measure $CH_3CN$ profiles. The error in a $CH_3CN$ vertical profile from the Level-2 research (L2r) product version 3.0.0 was estimated by both theoretical error analysis and compared with other

instrumental measurements. We estimated the systematic and random errors to be $\sim$ 5.8 ppt (7.8 %) and 25 ppt (60 %) for a single observation at 15.7 hPa, respectively, in the Tropics, where the $CH_3CN$ measurements are enhanced. The major source of systematic error was a pressure broadening, and its contribution to the total systematic error was approximately 60 % in the middle stratosphere (15.7–4.8 hPa). The random error decreased to less than 40 % after averaging 10 profiles in the pressure range of 28.8–1.6 hPa. The total error due to uncertainties in other molecular spectroscopic parameters was

comparable (2.8 ppt) to those of $CH_3CN$ spectroscopic parameters. We compared the SMILES $CH_3CN$ profiles with those of the Microwave Limb Sounder (MLS) on the Aura satellite (version 4.2). The SMILES $CH_3CN$ values were consistent with those from MLS within the standard deviation (1 $\sigma$) of the MLS observations. The difference between the SMILES and MLS $CH_3CN$ profiles increased with altitude and was within 20–35 ppt (20–260 %) at 15.7–1.6 hPa. We observed discrepancies of 5–10 ppt (10–30 %) between the SMILES $CH_3CN$ profiles observed by different spectrometers, so we do not recommend

merging SMILES $CH_3CN$ profiles derived from the different spectrometers. We found that SMILES $CH_3CN$ VMR in the upper stratosphere has a seasonal maximum in February, which is consistent with the fact that biomass burning events are highest from December–March.

## 1   Introduction

Air pollution derived from biomass burning (BB) has become a serious problem with population growth (Marlon et al., 2008).

BB events are important sources of various trace gases and particles in the atmosphere (Eagan et al., 1974; Crutzen et al., 1979). The study of atmospheric gas species associated with BB is significant because early estimates of pyrogenic emissions





suggested that some atmospheric pollutants from BB could be comparable to fossil fuel burning (Crutzen and Andreae, 1990; Seiler and Crutzen, 1980). These emissions could therefore, significantly affect the global atmosphere and its temperatures (Andreae, 1983).

Acetonitrile ($CH_3CN$) is a good tracer for BB as it is one of the dominant gases emitted during wildfire events (90–95 %) (Li et al., 2003). The mean lifetime of $CH_3CN$ in the atmosphere is about 6.6 months, with ocean uptake and the reaction with hydroxyl radicals (OHs) (Singh et al., 2003; de Gouw, 2003). Chemical loss of $CH_3CN$ with OH radicals occurs primarily in the stratosphere, whereas oceanic loss is dominant in the troposphere. Carbon monoxide (CO) is also a well-known BB tracer, but it only has an atmospheric lifetime of about 2 months in the free troposphere. CO is also emitted from some
anthropogenic sources, so $CH_3CN$ is not only longer-lived, but is also more specific to BB, and is therefore a better tracer. Arnold et al. (1978) first measured the presence of stratospheric $CH_3CN$ from the composition of positive ions using active chemical ionization mass spectrometry. $CH_3CN$ has also been detected using balloon-borne and airborne measurements in the lower stratosphere Knop and Arnold (1987); Schneider et al. (1997). More recently, satellite observations of $CH_3CN$ in the lower stratosphere have been measured using several satellite instruments, such as Microwave Limb Sounder (MLS)
onboard the UARS (Upper Atmosphere Research Satellite) (Barath et al., 1993), Atmospheric Chemistry Experiment Fourier Transform Spectrometer (ACE-FTS) onboard the Scisat-1 (Bernath, 2001), MLS onboard the Aura (Waters et al., 2006), and Superconducting Submillimeter-Wave Limb-Emission Sounder (SMILES) onboard the JEM (Japanese Experiment Module) of the International Space Station (ISS) (Kikuchi et al., 2010). Previous work reported the volume mixing ratio (VMR) of $CH_3CN$ mainly in the upper troposphere and lower stratosphere (UTLS) (Livesey et al., 2001, 2004; Harrison and Bernath,
2013). However, there are only a few reports of the VMR of $CH_3CN$ for the lower stratosphere to mesosphere.

Here, we derived vertical distribution profiles of $CH_3CN$ between the lower stratosphere and mesosphere from SMILES observations. We also performed a validation analysis comparing the results with Aura/MLS observation data.

## 2   SMILES $CH_3CN$ observations

The JEM/SMILES operated from October 12th 2009 until April 21st 2010 on the ISS (Kikuchi et al., 2010). The ISS has a
non sun-synchronous orbit and an inclination angle of 51.6° to the equator, which enables it to observe the atmosphere under various local solar times. The antenna field of view of the SMILES instrument was set to point in a 45° direction leftward from the ISS orbital motion. Low temperature system noise ($T_{sys} \sim 350$ K) was achieved using the four kelvin cooled submillimeter wave superconductive heterodyne receivers (Ochiai et al., 2011). This noise level is ten times lower than previous observations (Kikuchi et al., 2010). A summary of characteristics for SMILES observation is shown in Table 1.
The targeted $CH_3CN$ transition at 624.82 GHz for $(J, K) = (33, 3)$–$(33, 4)$ is allocated with a frequency region of Band-A (624.32–625.52 GHz) as shown in Fig 2. SMILES employed two Acousto Optical Spectrometers (AOSs) with a bandwidth of 1.2 GHz, which we denote as AOS1 and AOS2. The band configuration for AOS1 and AOS2 are summarized in Table 2. The date of observations made by AOS1 and AOS2 are shown in Fig. 1. The two AOSs detect Band-A, B, or C separately, enabling SMILES to observe two of the three bands simultaneously.



**Table 1.** SMILES characteristics.

| Parameter | Characteristics |
|---|---|
| Orbit | Non sun synchronous orbit |
| | $\sim 91$ min orbital period |
| Latitude coverage | $38^\circ$ S–$65^\circ$ N (nominal) |
| Integration time | 0.47 sec |
| Number of data | 1630 scan per day |
| Frequency range | 624.32–625.52 GHz (Band-A) |
| | 625.12–626.32 GHz (Band-B) |
| | 649.12–650.32 GHz (Band-C) |
| Receiver system | SIS mixers and HEMT amplifiers[†] |
| Spectrometers | Acousto Optical Spectrometers |
| | (AOS1 and AOS2) |
| Frequency resolution | 0.8 MHz |
| System noise temperature | $\sim 350$ K |

[†] SIS:Superconductor–insulator–superconductor mixer;

HEMT: High electron mobility transistor

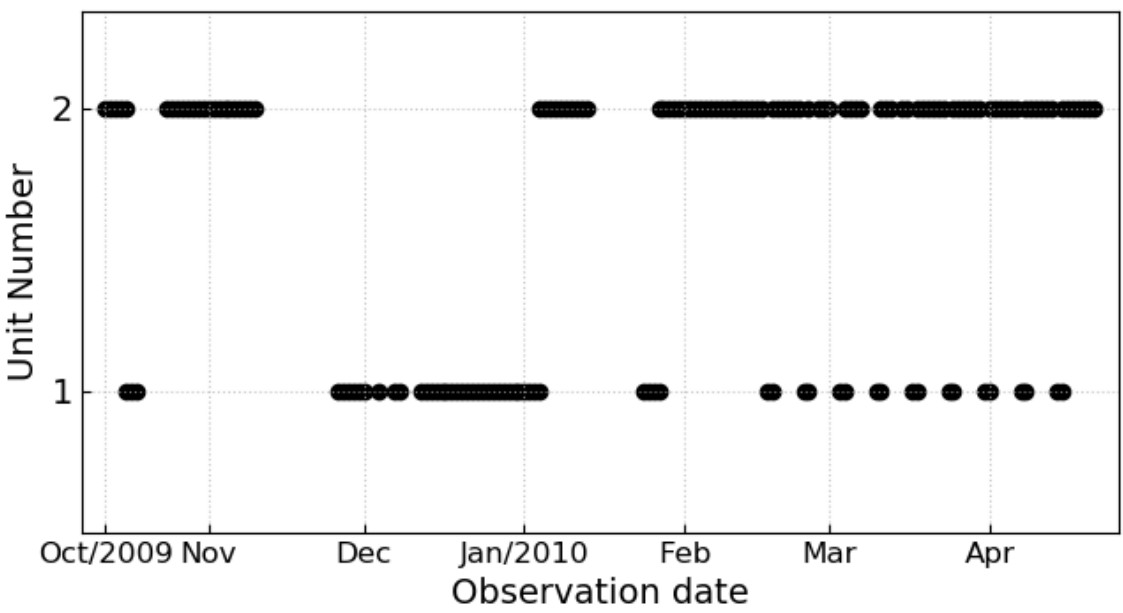

**Figure 1.** The distribution of AOS unit number for the SMILES $CH_3CN$ observation date.



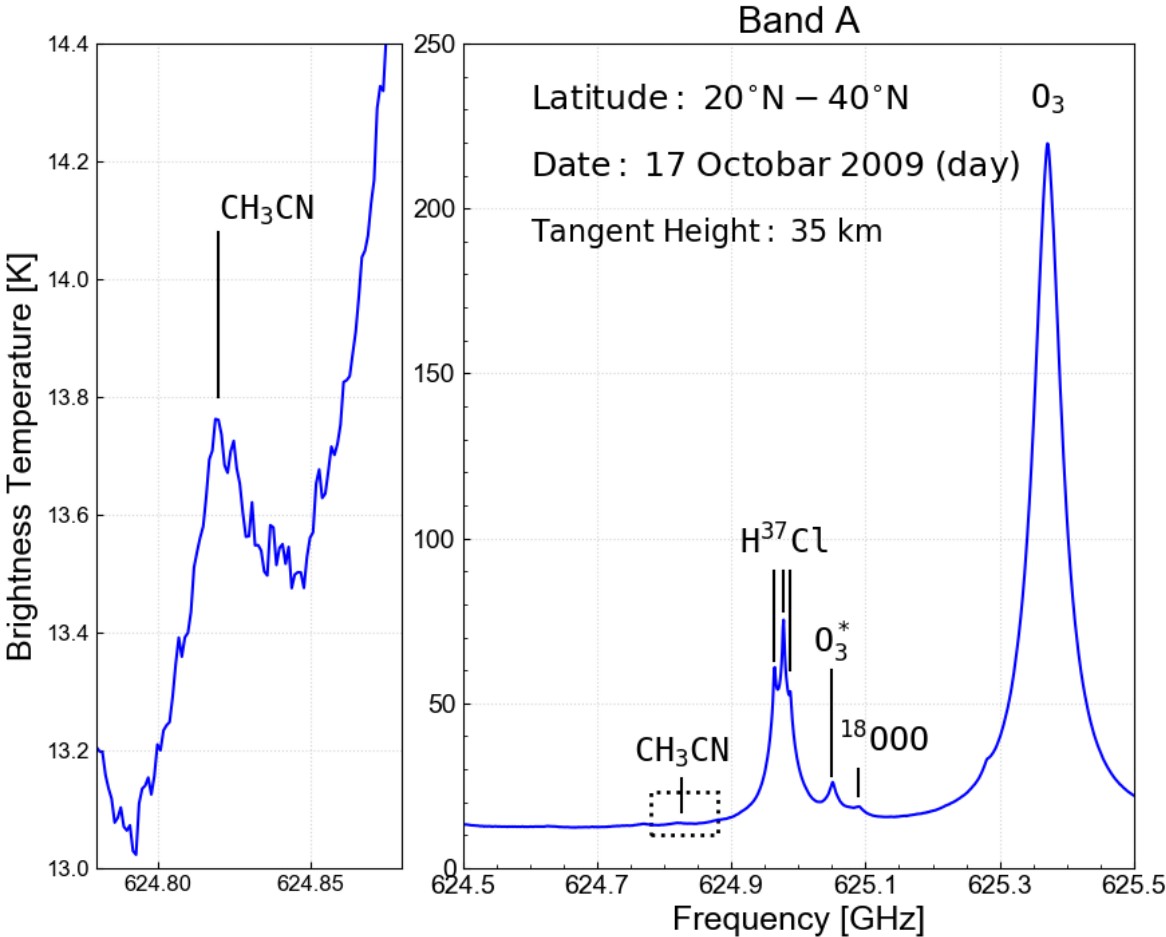

**Figure 2.** Example of SMILES spectrum (L1b ver. 008) of Band-A. 50 scans were accumulated in a tangent height of $35\pm2.5$ km over the daytime on October 17th 2009.

**Table 2.** Band configurations

| Band config. no. | AOS1 | AOS2 |
| --- | --- | --- |
| #1 | Band-A | Band-B |
| #2 | Band-C | Band-B |
| #3 | Band-C | Band-A |





The SMILES Level 2 research (L2r) product version 3.0.0 (v3.0.0) was used in this study. The $CH_3CN$ VMR profile was retrieved from the measurement spectra data of the Level-1b (L1b) version 008. Major improvements of the v3.0.0 from the previous version 2.1.5 were the AOS response function and a priori temperature profile. The details can be found in the JEM/SMILES L2r data product guideline (see *http://smiles.nict.go.jp/pub/data/index.html*). The optimal estimation method (OEM) was used for the retrieval processing. The OEM leads to the maximum a posteriori solution, which minimizes the value

of $\chi^2$ described below.

$$\chi^2 = [\mathbf{y} - \mathbf{F}(\mathbf{x}, \mathbf{b})]^T \mathbf{S}_y^{-1}[\mathbf{y} - \mathbf{F}(\mathbf{x}, \mathbf{b})] + [\mathbf{x}_a - \mathbf{x}]^T \mathbf{S}_a^{-1}[\mathbf{x}_a - \mathbf{x}] \tag{1}$$

where $\mathbf{F}(\mathbf{x}, \mathbf{b})$ is the forward model depending on $\mathbf{x}$ state vector and on the known model parameters $\mathbf{b}$, $\mathbf{S}_y^{-1}$ the measurement covariance matrix, $\mathbf{x}_a$ the a priori state of $\mathbf{x}$, and $\mathbf{S}_a$ the a priori covariance matrix. Detailed retrieval algorithm of L2r product can be found in Baron et al. (2011) and Sato et al. (2012).

Quality of the retrieval processing was quantified by the chi-square statics, or goodness of the fit (Eq. 1), and the measurement response ($\mathbf{m}$) defined as,

$$\mathbf{m}[i] = \sum_j |\mathbf{A}[i,j]| \tag{2}$$

$$\mathbf{A} = \frac{\partial \hat{\mathbf{x}}}{\partial \mathbf{x}} = \mathbf{DK} \tag{3}$$


$$\mathbf{D} = \frac{\partial \hat{\mathbf{x}}}{\partial \mathbf{y}} = (\mathbf{K}^T \mathbf{S}_y^{-1} \mathbf{K} + \mathbf{S}_a^{-1})^{-1} \mathbf{K}^T \mathbf{S}_y^{-1} \tag{4}$$

$$\mathbf{K} = \frac{\partial \mathbf{y}}{\partial \mathbf{x}} \tag{5}$$

where $\hat{\mathbf{x}}$ is the solution of the retrieval, $\mathbf{A}$ is the averaging kernel, $\mathbf{D}$ is the contribution function, and $\mathbf{K}$ is the weighting function.

$\mathbf{m}$, $\mathbf{A}$ and $\mathbf{D}$ were derived using $\mathbf{K}$ (Urban et al., 2004). Details on $\mathbf{m}$ are explained by Sato et al. (2014). The $\chi^2$ of $CH_3CN$ for v3.0.0 had a range of 0.4–0.6. In cases where the measurement response was low, information was retrieved from the a priori state. Here, the data selection thresholds of $\chi^2$ and measurement response were set to be $\chi^2 < 0.6$ and $\mathbf{m} > 0.80$, respectively.

Figure 3 shows an example of the retrieval results from a single spectral scan on November 4th 2009, in the Tropics at latitude less than 20°, including the retrieved $CH_3CN$ vertical profile, averaging kernel and vertical resolution. The vertical resolution

was defined as the full width at half maximum (FWHM) for each row of the averaging kernel matrix. The measurement response of retrieved $CH_3CN$, shown as a black solid line in the middle panel of Fig. 3, is the sum of elements from the averaging kernel at each altitude. The measurement response was almost one from 30 to 55 km, with a vertical resolution of 7–15 km, decreasing with altitude.



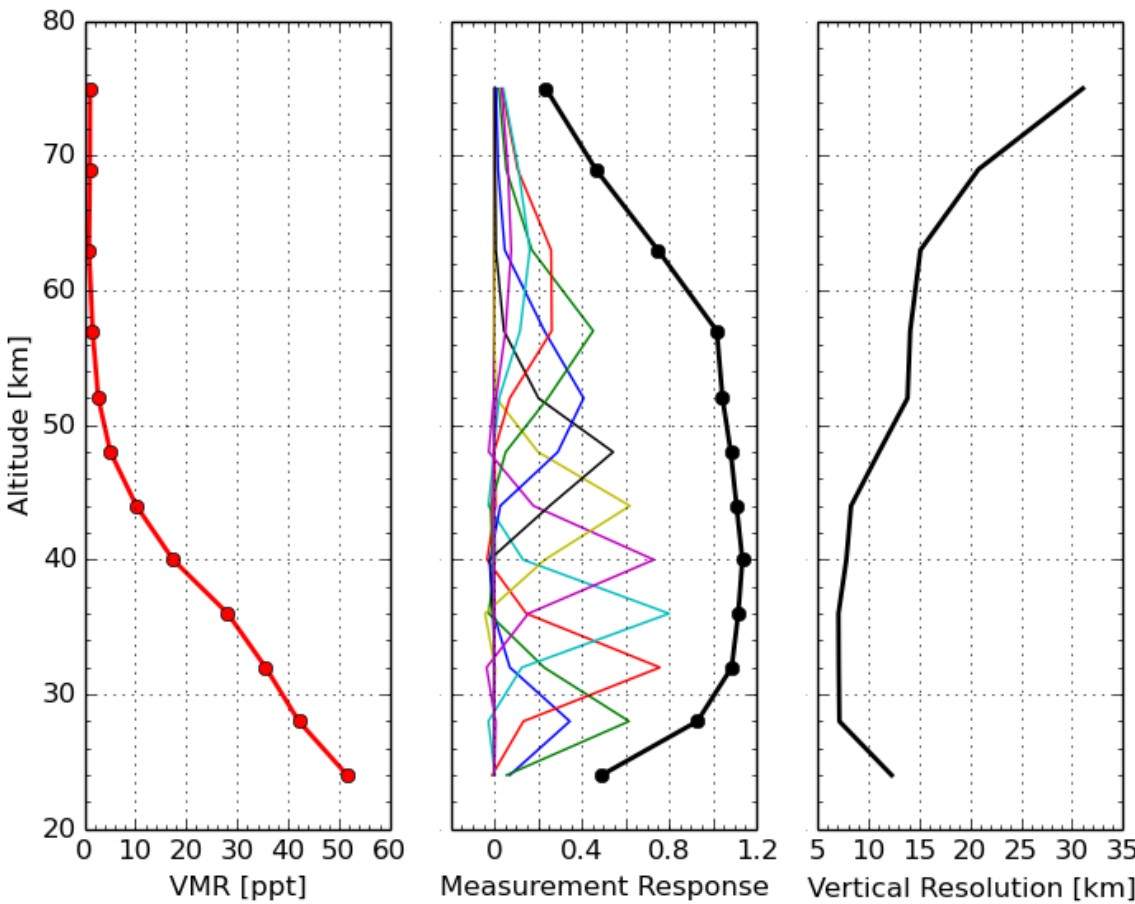

**Figure 3.** (*Left*) Vertical profile of $CH_3CN$ retrieved from a single spectral scan on November 4th 2009, in the Tropics at latitude less than $20°$. (*Middle*) The averaging kernel by altitude, for each measurement (color line), and the measurement response (solid black line). (*Right*) The vertical resolution of the profile.





## 3 Theoretical error analysis

We theoretically estimated the error in deriving $CH_3CN$ profiles from SMILES observations by perturbing the model parameters in a forward model (Sato et al., 2012; Kasai et al., 2013; Sagawa et al., 2013). We used a typical $CH_3CN$ profile derived using observations from the Tropics, where BB (a major source of $CH_3CN$) frequently occurs. The total error ($\mathbf{E}_{total}$) is given by

$$\mathbf{E}_{total}[i] = \sqrt{\mathbf{E}_n^2[i] + \mathbf{E}_s^2[i] + \mathbf{E}_p^2[i]}, \tag{6}$$

where $\mathbf{E}_n$ is the error due to spectral noise, $\mathbf{E}_s$ is the smoothing error, and $\mathbf{E}_p$ is the model parameter error. The error due to the spectral calibration was ignored in this study, because the L1b data was updated in this version, and the error due to the spectral calibration was not significant in previous SMILES error analyses (e.g. Sato et al., 2012).

Error $\mathbf{E}_n$ and $\mathbf{E}_s$ was calculated by

$$\mathbf{E}_n[i] = \sqrt{\mathbf{S}_n[i,i]}, \tag{7}$$

where

$$\mathbf{S}_n = \mathbf{D}\mathbf{S}_y\mathbf{D}^T, \tag{8}$$

and

$$\mathbf{E}_s[i] = \sqrt{\mathbf{S}_s[i,i]}, \tag{9}$$

where

$$\mathbf{S}_s = (\mathbf{A} - \mathbf{U})\mathbf{S}_a(\mathbf{A} - \mathbf{U})^T. \tag{10}$$

Here, $\mathbf{S}_n$ and $\mathbf{S}_s$ are the error covariance matrices for measurement noise and the errors from $\mathbf{S}_a$, respectively. $\mathbf{U}$ is the unit matrix.

The model parameter error $\mathbf{E}_p$ includes errors caused by uncertainties in the parameters used in both the forward and inversion calculations. Error sources for the model parameters are summarized in Table 3. Error related to each of the individual
model parameters was calculated using the perturbation method following Sato et al. (2012). The total error $\mathbf{E}_p$ for all of the parameters was calculated using the root sum square of the individual errors.

Figure 4 shows the estimated systematic errors.The left panel (a) shows the uncertainties in the AOS response function ("AOS"), the antenna beam pattern ("Antenna"), the spectral line strength ("Strength"), the air pressure broadening coefficient ("$\gamma$"), its temperature dependence ("$n$"), and their root sum square ("Total"). The largest error source ($\sim 2$ ppt (5 %) was
from the air pressure broadening coefficient ("$\gamma$") across the entire pressure range, followed by line intensity ("Strength") and temperature dependence of air pressure broadening coefficient("$n$") ($\simeq 1.5$ ppt). The error from spectroscopic parameters was more significant than that from instrumental functions.



**Table 3.** Potential error sources.

| Error source | Uncertainty |
|---|---|
| Spectroscopic parameter of $CH_3CN$ | |
| Line intensity (Strength) | 1 % |
| Air pressure broadening ($\gamma$) | 3 % |
| Temperature dependence of $\gamma$ ($n$) | 10 % |
| Instrumental functions | |
| AOS response function (AOS) | 10 % |
| Antenna scan (Antenna) | 2 % |
| Impact from other species | |
| $H^{37}Cl$ air pressure broadening ($H^{37}Cl\gamma$) | 3 % |
| Temperature dependence of $H^{37}Cl\gamma$ ($H^{37}Cln$) | 10 % |
| $O_3$ air pressure broadening ($O^3\gamma$) | 3 % |
| $O_3$ temperature dependence of $O^3\gamma$ ($O^3n$) | 10 % |

In Band-A, $O_3$ and $H^{37}Cl$ are observed near the $CH_3CN$ transition (See Fig. 2). The spectral shape of $O_3$ and $H^{37}Cl$ should therefore influence the retrieval of the $CH_3CN$ VMR profiles. To estimate the influence from the other spectral lines, error due

to the spectroscopic parameters $\gamma$ and its temperature dependence $n$ of the $O_3$ and $H^{37}CL$ lines were also calculated. $\gamma$ and temperature dependence of $\gamma$ were perturbed for each species, and are expressed as "$O_3\gamma$", "$O_3n$", "$H^{37}Cl\gamma$" and "$H^{37}Cln$". As shown in Fig. 4 (**b**), "$H^{37}Cl\gamma$" is the largest error source, whose maximum absolute difference was 1.1 ppt. Error analyses completed for $O_3$ and ClO demonstrated that error caused by other molecular spectral lines was negligible as they have high, isolated line strengths (Sato et al., 2012; Sagawa et al., 2013; Kasai et al., 2013). In the case of $CH_3CN$ retrieval, however, the

total error caused by uncertainties in other molecular spectroscopic parameters was comparable to the error caused by $CH_3CN$ spectroscopic parameters. The errors due to $H^{37}Cl$ was larger than that from $O_3$ at each pressure level.

The measurement noise and smoothing error from a single scan are shown in the Fig. 5 (**a**). These errors are considered random error for a $CH_3CN$ profile. SMILES $CH_3CN$ total error consists of both the systematic and random error. Figure 5 (**b**) shows the total systematic error, the random and total error averaged by the number of profile ($N$ = 1, 10 and 100). The

random error was larger than the systematic error from a single scan. However, the random error averaged by 100 profiles was comparable to the systematic error, except for the highest systematic error, which was found at a pressure level of about 1 hPa.

## 4 Comparison with Aura/MLS

In this section, we compare SMILES $CH_3CN$ observations with Aura/MLS observations and discuss the validity of SMILES observations.

Figure 6 shows (**a**) a $CH_3CN$ vertical profile observed by AOS1 and AOS2, (**b** and **c**) the absolute and relative differences between AOS1 and AOS2 observed in Equatorial regions (20° S–20° N) from March until April 2010, when AOS1 and AOS2



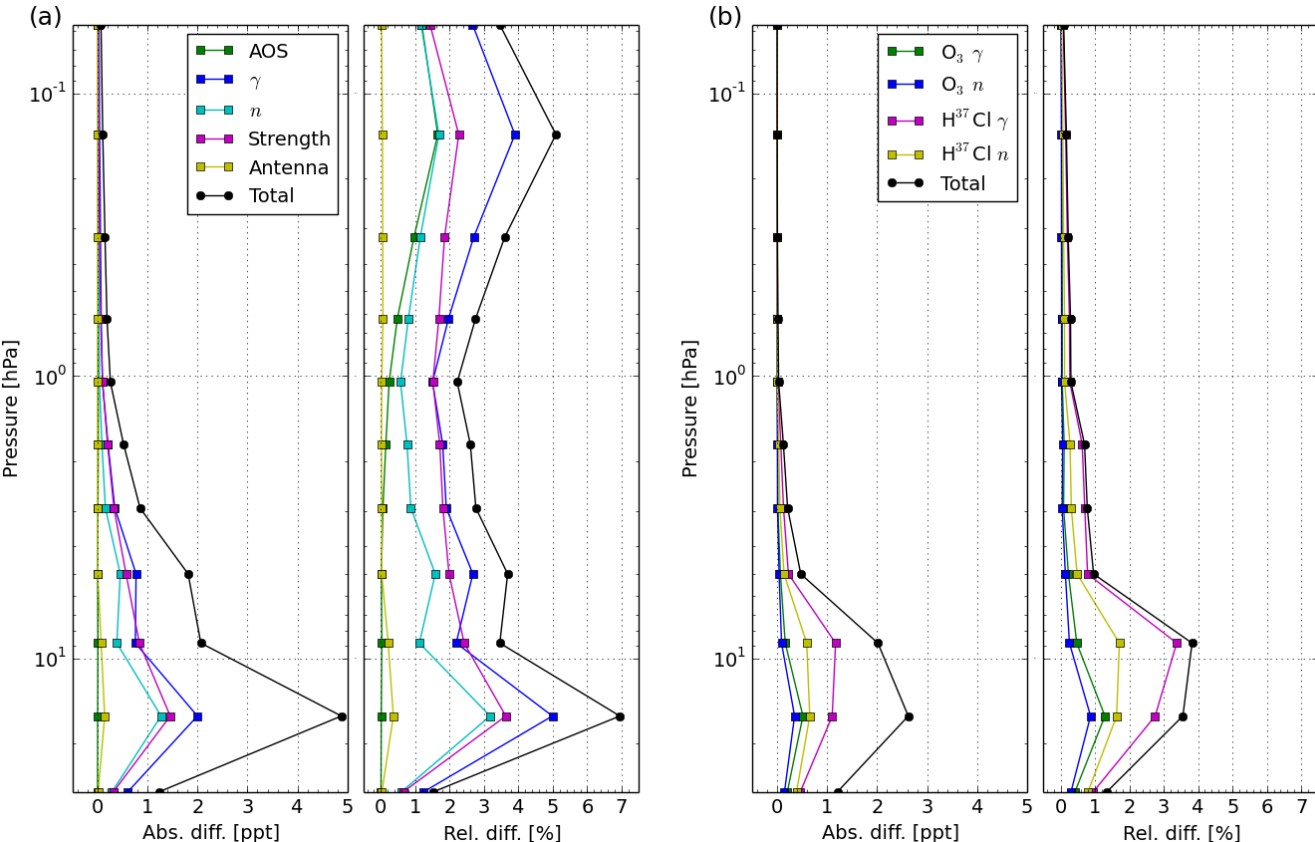

**Figure 4. (a)** Summary of absolute and relative differences derived from systematic errors of $CH_3CN$ retrieval caused by uncertainties in the spectral parameters and **(b)** instrumental functions derived from single scan spectrum observed on November 4th 2009, in the Tropics, as shown in Fig. 3. The black line in the middle panel indicates the total error calculated by root-sum-square of all assumed error sources.

were alternating, at a ratio of 1:3. The error bars shown in left panel of Fig. 6 are standard deviations ($1\sigma$) of the $CH_3CN$ VMRs observations retrieved at SMILES pressure grids for AOS1 (blue) and AOS2 (green). The relative difference between AOS1 and AOS2 is approximately 12 ppt (30 %) with the maximum at 15.7 hPa, indicating that the difference between the two
AOSs is due to sensitivity differences. The difference between the two decreases down to less than 10 % at an upper altitudes than 4.8 hPa.

We also investigated seasonal variation of SMILES $CH_3CN$ observations for each altitude grid as shown in Fig. 7. This figure shows daily scatter plots and daily averages for AOS1 (red shaded) and AOS2 (blue shaded) observations. The red circles and bars represent the daily mean values and $1\sigma$ standards deviation, when more than one hundred observation points
were obtained in one day. Like in Fig. 1, at lower altitudes (28 km to 36 km) the difference between the two AOSs observations was significantly larger, especially from December until the beginning of January. However, in the upper stratosphere there was no difference between the two AOS observations, and the standard deviations decreased with altitude. In terms of seasonality,



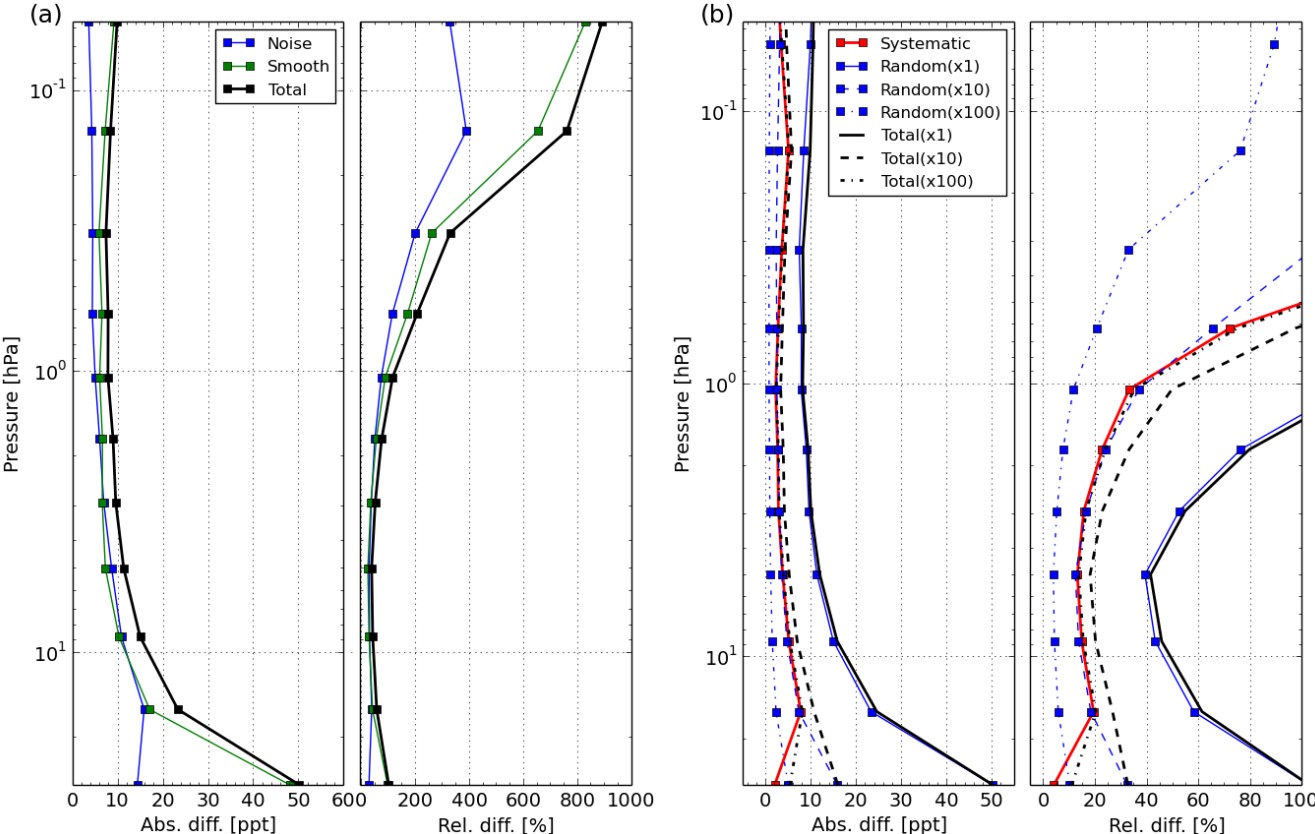

**Figure 5. (a)** Summary of absolute and relative differences derived from random errors of $CH_3CN$ retrieved from a SMILES single scan observation as shown in Fig. 4. **(b)** Summary of absolute and relative differences derived from random (blue), systematic (red), and total (black) errors in the SMILES $CH_3CN$ retrieval for the averaging of $N$ profiles ($N = 1, 10,100$) The number in the legend is the accumulating profile number.

$CH_3CN$ levels peaked in February, and can be seen from approximately 40 km to 52 km where the difference between the two AOSs can be negligible.

## 4.1 Comparison with Aura/MLS v4.2 data

We investigated the difference of $CH_3CN$ VMRs between SMILES and MLS observations. We set the data quality thresholds and the coincidence selection criteria for the SMILES and MLS observations, as summarized in Table 4. The MLS data quality criteria was based on the MLS v4.2 Level-2 data quality and description document.

The geolocation and measurement time criteria were determined as follows;

– the distance of measurement location within 300 km;

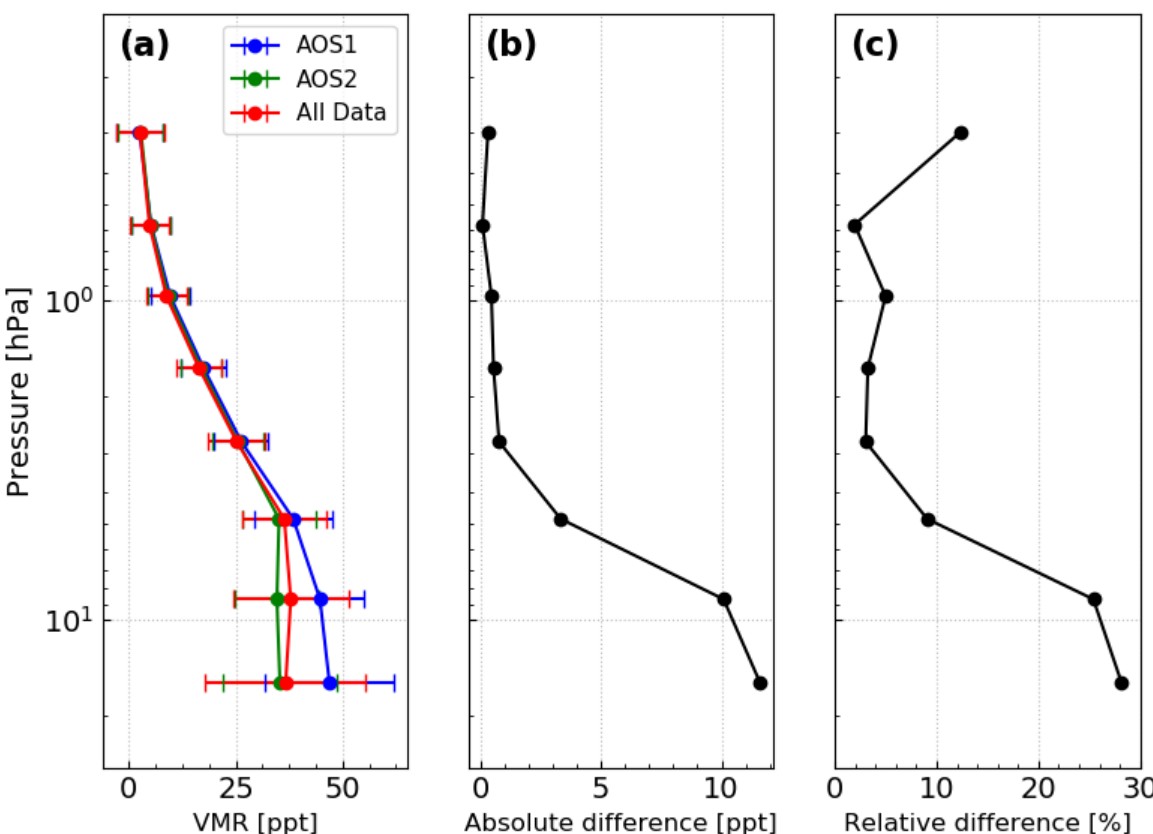

**Figure 6.** (a) Vertical profiles of $CH_3CN$ from AOS1, AOS2 and the sum of AOS1 and AOS2 in the Equatorial region from $20°S$ to $20°N$, from March until April 2010. Each line indicates the averaged VMR from AOS1 observations (blue line), AOS2 observations (green line) and the sum of AOS1 and AOS2 observations (red line). (b) The absolute difference between AOS1 and AOS2 ($AOS1 - AOS2$). (c) The relative difference between AOS1 and AOS2 ($(AOS1 - AOS2)/M$ when $M$ is $(AOS1 + AOS2)/2$).

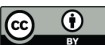

**Figure 7.** Daily scatter and average plots for retrieved $CH_3CN$ observations at each altitude (28–52 km) in the Equatorial region ($20°$S–$20°$N). Solid red lines indicate filtered mean values. Error bar indicates $1\sigma$ standard deviation of the moving average. Red (blue) shaded areas represent the date observed by AOS1 (AOS2).



**Table 4.** Data quality criteria for SMILES and MLS

| Data products | Quality threshold |
| --- | --- |
| SMILES v3.0.0 | Measurement response > 0.80 |
| | Goodness of fit ($\chi^2$) < 0.6 |
| | Field-of-view = 0 |
| MLS v4.2 | Quality > 1.40 |
| | Convergence < 1.05 |
| | Status = 0 |

- difference in the measurement time within 6 hour.

We investigated the diurnal variation of SMILES $CH_3CN$ observations at several altitudes (32 km, 40 km, and 48 km) for AOS1 and AOS2 individual observational periods, and confirmed that there is no diurnal variation for stratospheric $CH_3CN$ observations.

Figure 8 shows the distribution of coincident points satisfying these criteria at 8.6 hPa. The interpolation of VMRs was done using a linear interpolation with respect to the logarithm pressure levels. There are on average 10 coincident points in each bin at this pressure level and the total coincident data number was 17910.

For the comparison between SMILES and MLS observations, the mean absolute difference, $\Delta_{abs}$, and relative difference, $\Delta_{rel}$, at the pressure levels, $p$, between coincident $CH_3CN$ profiles of the two observations were calculated as follows,

$$\Delta_{abs} = \frac{1}{N(p)} \sum_{i=1}^{N(p)} \{x_s(p) - x_m(p)\}, \tag{11}$$

$$\Delta_{rel} = \frac{1}{N(p)} \sum_{i=1}^{N(p)} \frac{\{x_s(p) - x_m(p)\}}{\overline{x}(p)}, \tag{12}$$

where $N(p)$ is the number of coincidences at $p$, $x_s(p)$ and $x_m(p)$ are the VMRs at $p$ for SMILES and MLS observations, and the reference ($\overline{x}_p$) is $\overline{x}_p = \frac{1}{2}(x_s(p) + x_m(p))$.

### 4.1.1 Aura/MLS v4.2

The MLS has been onboard the Aura satellite since 2004 and has observed $CH_3CN$ levels from the lower to upper stratosphere. This satellite was launched in sun-synchronous orbit with an equator-crossing time 13:45 (ascending) and 01:45 (descending). The daily MLS measurements gives 82° S to 82° N latitude coverage. The MLS measures temperature and trace gases ($O_3$, CO, $H_2O$, $HNO_3$, $CH_3CN$ etc.) using thermal emission data from the atmosphere. The $CH_3CN$ VMR values were retrieved from the MLS observation data using the optimal estimation method. Details on the retrieval algorithm was described in Livesey et al. (2006). The MLS uses spectral bands of 118, 190, 240 and 640 GHz and 2.5 THz, observing $CH_3CN$ from 640 GHz



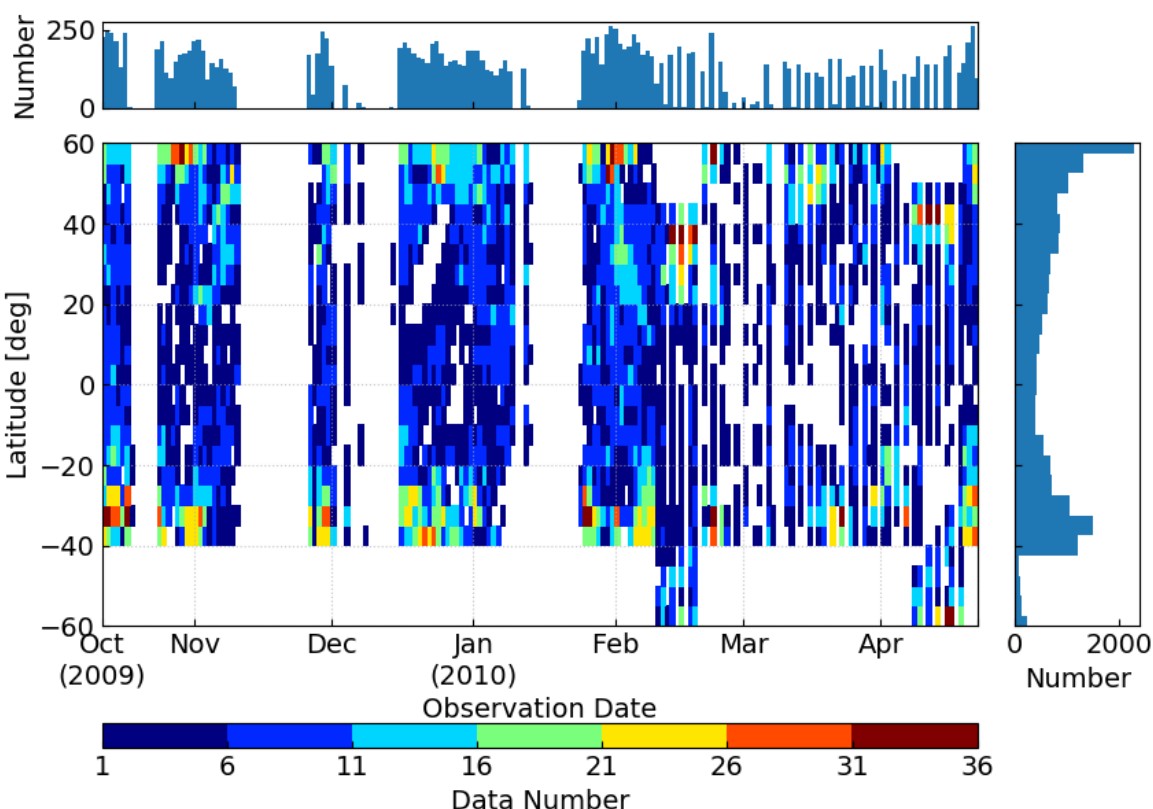

**Figure 8.** Distribution of the data meeting the criteria between October 12th 2009 and April 21st 2010, at 8.6 hPa. Observation date and latitude bins are 1 day and 3°.

spectral regions (Waters et al., 2006). MLS Level-2 $CH_3CN$ profiles were observed in 640 GHz spectral regions. Although the pressure range of a retrieved MLS $CH_3CN$ is 147 to 0.001 hPa, the pressure range of $CH_3CN$ version 4.2.0 is 46–1.0 hPa (Livesey et al., 2006).

**4.1.2 Result of comparisons**

Figure 9 shows the vertical profile, the absolute differences and the relative differences between SMILES AOS1/AOS2 and MLS $CH_3CN$ observations. The left panel in Fig. 9 indicates good agreement among the three observations from 15.7 hPa to 4.8 hPa. Across the range of the pressure levels, the absolute difference and the relative difference among the three observations were -15–25 ppt and 20–80 %, respectively. The difference between SMILES and MLS observations becomes larger with 180 altitude, from a pressure level of 8.6 hPa. Overall, the variance of SMILES observations is smaller than that of MLS as SMILES





$T_{sys}$ was much smaller than that of MLS, indicating that SMILES has an advantage in the upper stratosphere. SMILES was also able to observe $CH_3CN$ VMR in the upper stratosphere with a much lower uncertainty of $\sim$20 ppt although the uncertainty of MLS $CH_3CN$ VMR was approximately 100 ppt in the altitude. The differences of the $CH_3CN$ VMR observed by two AOSs was sufficiently small in comparison with the difference between SMILES and MLS observations. Theoretical systematic error

(blue broken lines in the middle panel) derived in Sect. 3 was less than the differences between SMILES and MLS observations, except at 8.9 hPa.

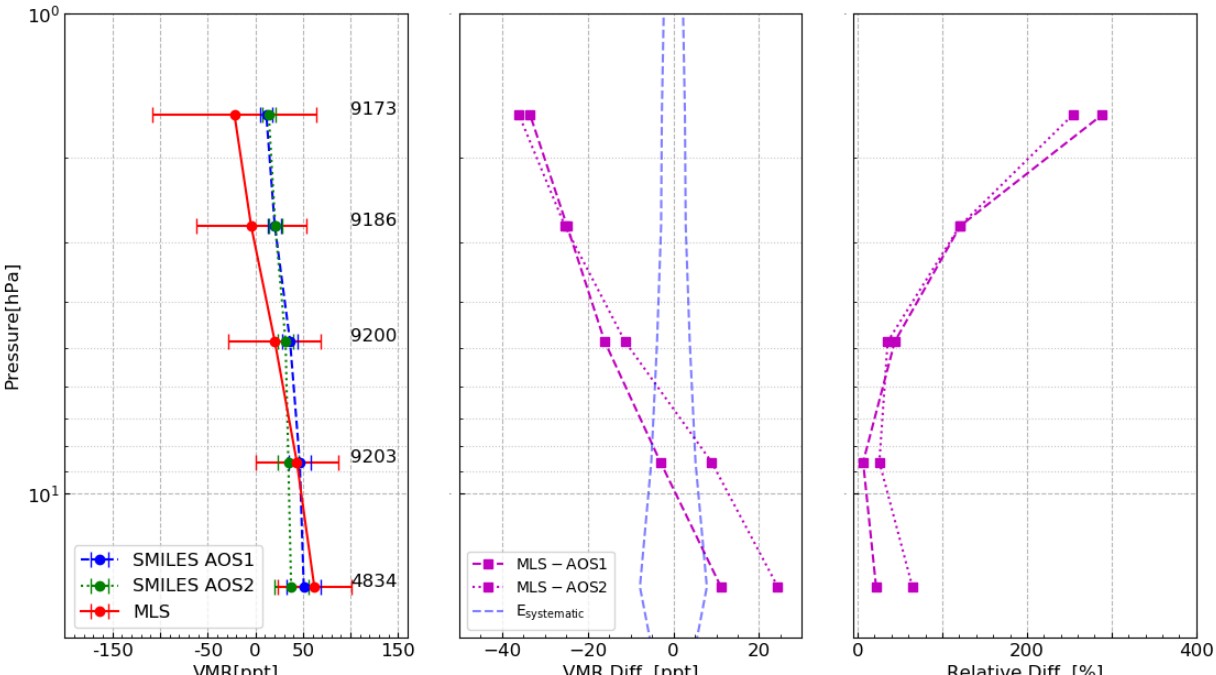

**Figure 9.** (*Left*) Mean $CH_3CN$ VMR values and the standard deviations for SMILES and MLS measurements. The blue and green lines represent the SMILES VMR observed by AOS1 and AOS2, respectively. The red line represents the MLS VMR. Error bars indicate $1\sigma$ standard deviation for each dataset. The number of coincident data are displayed at each point. (*Middle*) The absolute difference in the mean $CH_3CN$ VMR values between SMILES AOS1/AOS2 and MLS observations is calculated by Eq. 11. Blue broken lines indicate systematic errors theoretically derived in Sect. 3. (*Right*) The relative differences of $CH_3CN$ levels observed between SMILES and MLS methods is calculated by Eq. 12

We also investigated latitudinal and seasonal variation between the two observation methods. Figure 10 shows the seasonal variation of SMILES and MLS $CH_3CN$ observations, and the absolute differences for each pressure level at coincident points, as a function of latitude. The left column represents SMILES $CH_3CN$ VMR in units of ppt which were separated into two AOSs

observations. The middle column represents MLS $CH_3CN$ VMR, and the right column represents the absolute differences between SMILES and MLS observations. At lower altitudes of 15.7 hPa and 8.6 hPa, SMILES observations were overestimated





compared to MLS observations, while at upper levels (4.8 hPa∼) it was underestimated up to 40 ppt. At every pressure level, SMILES CH₃CN VMRs were higher in the Tropics (20° S ∼ 20° N).However, in the case of MLS observations at higher pressure levels, the number of coincident points significantly decreased. CH₃CN levels in the Tropics were ambiguous at

pressure levels above from 4.8 hPa, indicating that in the upper stratosphere it is hard to observe latitudinal and seasonal trends of CH₃CN, due to the large uncertainty of MLS observations. At pressure levels above 4.8 hPa, SMILES CH₃CN observation in February are comparable with the other periods. This result indicates that CH₃CN in the upper stratosphere reaches its seasonal maximum in February, which is consistent with our understanding that most biomass burning occurs from December–March.

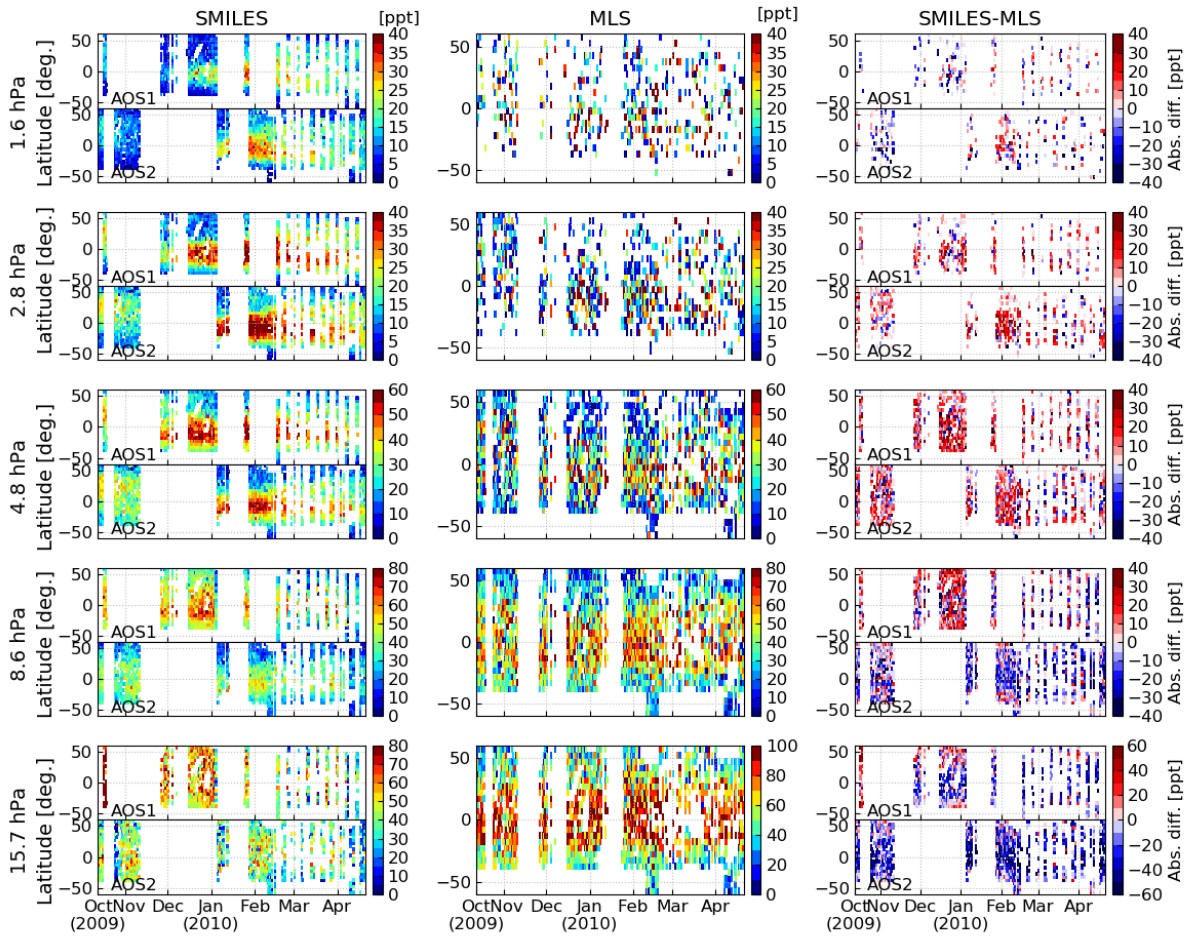

**Figure 10.** Seasonal variation of SMILES and MLS CH₃CN observations and the absolute difference for each pressure level, as a function of latitude. Observation date and latitude bins are 1 day and 5° grid

.



## 5   Conclusions

Our analysis demonstrates the validity of using SMILES observations to measure $CH_3CN$ profiles. We were able to successfully derive vertical profiles and observe seasonal variation of $CH_3CN$ in the stratosphere, using SMILES observations. In doing so, this study is the first to show results observed from satellite instruments of $CH_3CN$ VMR from the upper stratosphere to lower mesosphere with much lower uncertainty of 20 ppt. Error analysis showed that random error was the dominant source of uncertainty (around 25 ppt at 15.7 hPa) in the measurement altitude range. The uncertainty of air pressure broadening was the dominant systematic error source, with a maximum difference of 2.0 ppt (5 %). The random error from single scan spectrum was more than two times larger than systematic error at 15.7 hPa, while the random error averaged with 100 spectra was comparable to systematic error. SMILES and Aura/MLS observations were in agreement in the stratosphere from 15.7 hPa to 4.8 hPa. At upper pressure levels the difference between the two observations increased up to 35 ppt (260 %) because of greater uncertainty of Aura/MLS observations, and because $CH_3CN$ levels were at their seasonal maximum. These seasonal maximum $CH_3CN$ levels measured in February are consistent with the yearly high period of BB events from December–March. The theoretical systematic error and the relative difference of the SMILES measurements compared to MLS measurements were 10 ppt and 35 ppt at altitudes between 15.7 hPa and 1.6 hPa (28–44 km). Furthermore, the two AOSs show comparable errors (∼10 ppt) at 0.93 hPa to 0.29 hPa (approximately 48–56 km) and at lower pressure levels, implying the reliability of SMILES $CH_3CN$ observations.

*Data availability.* The SMILES data is available at http://smiles.nict.go.jp/pub/data/index.html. The MLS data is available at https://mls.jpl.nasa.gov/data/.

*Author contributions.* TF designed the study and performed the analysis. YK designed the study and provided the SMILES data. TOS provided the retrieval code and contributed to data analysis and interpretation. TY, SN, YU, and KT contributed to data analysis and reviewed the manuscript. NY supervised and reviewed the manuscript. TF wrote the manuscript with contributions from all coauthors.

*Competing interests.* The authors declare that no competing interests are present.

*Acknowledgements.* This work was conducted as part of a research project titled 'R&D to Expand Radio Frequency Resources', which is supported by the Japanese Government through the Ministry of Internal Affairs and Communications (0155-0285, 0155-0093). We deeply appreciate Prof. Hideo Sagawa (Kyoto Sangyo University) for developing the SMILES L2r product v3.0.0. We are grateful to Atsushi Hirakawa and Kota Kuribayashi for supporting this study We are grateful to Nathaniel Livesey for providing the MLS version 4.2.0 product.



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
