# Peer review of "Validation of acetonitrile (CH3CN) measurements in the stratosphere and lower mesosphere from the SMILES instrument on the International Space Station"

_Atmospheric Measurement Techniques, 2019_

## Referee Comment (RC1) · Anonymous Referee #1 · 4 Oct 2019

This article describes retrievals of CH3CN from the SMILES instrument on board the international space station and performs a validation through comparison with results from the MLS instrument.

The level of English in the manuscript could use some work, starting with the title. It is readable, but the phrasing is occasionally not quite correct. There are too many instances for me to provide line-by-line corrections.

I have some issues with the background discussion on CH3CN. While it is all technically

correct that the molecule is associated with biomass burning and pollution and can be used as a tracer for biomass burning events, most biomass burning activity occurs in the troposphere but the measurements reported here are in the stratosphere. Some of the discussion is not entirely relevant to what they are measuring.

They report the observation of a seasonal maximum of CH3CN in the upper stratosphere in February supposedly resulting from a peak in biomass burning during the time period from December to March. Age of air in the upper stratosphere is the order of a few years, but they appear to be suggesting they are measuring enhanced CH3CN in the upper stratosphere almost immediately. It should take a few years for young tropospheric air containing enhanced CH3CN to make its way to that atmospheric region. Intense fires can inject biomass burning products directly into the stratosphere (pyroconvection), but I don't think that is what they propose is happening here. CH3CN levels would be anomalously high in that case, and I expect the effect would not extend into the upper stratosphere. It would be confined to the lower stratosphere. The authors may want to rethink their interpretation taking transport times to the upper stratosphere into consideration.

In Figure 7, for altitudes below 40 km, the systematic high bias of results from AOS1 relative to results from AOS2 are quite evident. The authors make the statement that the differences between the AOSs is due to "sensitivity differences." I do not know what that means. The phrasing would suggest something like signal-to-noise ratio, but I would expect that to yield increased variability and not a systematic offset, unless perhaps the retrieval is being "pulled" more toward the a priori in the optimal estimation analysis because of the reduced signal-to-noise ratio. Is it possible to provide a few extra words to explain what is meant by sensitivity differences? It might be instructive to see the signal from the two different AOSs under similar measurement conditions, to see why one is yielding a larger VMR than the other.

In Figure 9, the top VMR two points from MLS are negative, which contributes to the strong magnification of the discrepancies between the two instruments in this altitude

region. Negative VMRs have no physical meaning. You should probably point this out when discussing the discrepancies. This is an item in your favor.

How was the a priori state constructed? If MLS results went into generating the a priori used in the optimal estimation analysis, extra care might be required in using comparisons to MLS as part of a validation effort.

I feel that the description of the CH3CN retrievals was fine, and it looks like a good data product, but I was not convinced that their interpretation of the results (biomass burning on the surface yielding an almost immediate enhancement of CH3CN in the upper stratosphere) was correct.

---

## Referee Comment (RC2) · Hugh C. Pumphrey (Referee) · 31 Oct 2019

**1 General Comments**

This paper introduces a new measurement of CH$_3$CN and assesses its quality. The paper is generally well written and produced; it should be published subject to minor corrections. The standard of written English is generally good and it is nearly always clear what the authors mean. I identify a few exceptions below.

[Figure]

**2 Specific comments**

- Page 2 line 25: "Acetonitrile [. . . ] is one of the dominant gases emitted during wildfire events (90%-95%)". This statement is either wrong or confusing as wildfires emit far more CO and $CO_2$ than they do $CH_3CN$. What I suspect that the authors mean is that 90%-95% of $CH_3CN$ comes from wildfires. If that is what the authors mean, they should say so explicitly.

- Page 2 line 27:". . . ocean uptake and the reaction with hydroxyl radicals (OH)([. . . refs . . . ])." This sentence appears to be missing some words at the end. Maybe it should read ". . . ocean uptake and the reaction with hydroxyl radicals (OH)([. . . refs . . . ]) being the main loss mechanisms."

- Page 5 line 58 "maximum a posteriori solution" should be "maximum a posteriori probability solution". When mentioning the MAP solution it is surely obligatory to cite Rodgers (2000).

- Page 16, Figure 10. I struggle to interpret this figure. This is partly because the individual panels are rather small; I do not know what might be done about this as the arrangement of the figure is useful.

  The MLS data appear to have a lot of gaps in; the number of gaps increases with altitude. Now, it is clear that some of the gaps are there because the authors have chosen to show MLS data only for times when there is also SMILES data available. I would suggest that for the middle column of data they might want to show MLS data for all days, so that the eye can more easily pick out the patterns in the data.

  The actual MLS data do not become more sparse with altitude as the figure suggests: I just plotted them up to check. However, they **do** become negative on average, as can be seen in Figure 9 (left). I would suggest that the authors use a colour scale which spans the range of both the MLS and SMILES data. Clearly,

the MLS data are wrong in the sense that the atmosphere can not contain less than none of a constituent. But the time-latitude dependence of the MLS data is actually rather similar to that of the SMILES data — it would be nice if the plot could bring that out.

I am pleased to see a sequential colour scale used for the actual quantities and a diverging one for the SMILES-MLS differences. Less satisfactory is the choice of sequential colour scale; the authors have used the notorious "jet" scale, or something very like it. (See https://hughpumphrey.wordpress.com/2017/06/29/colours-for-contours/ for some thoughts and some useful links.) They might want to consider whether a scale other than "jet" might be appropriate in this figure.

- Page 16 Lines 194-196: The time-latitude dependence of the MLS data is actually quite clear and easy to see as long as you plot the data up with a colour scale that goes into the negative. Clearly, the MLS data have a negative bias of over 100% in the upper stratosphere, but this does not, of itself, prevent the seasonal behaviour being observed.

- Page 16 Lines 198-199 and page 17 lines 210-211: It is not at all clear to me why there should be a connection between the maximum observed at 1 hPa – 5 hPa in February and the timing of the biomass-burning season. Tropospheric source gases such as $CH_3CN$ enter the stratosphere via the tropical tropopause and take over a year to ascend from 100 hPa (16 km) to 10 hPa (32 km). This "tape recorder" effect was first observed in water vapour (Mote et al., 1996) and subseqently in HCN (Pumphrey et al., 2008, 2018) among other species. Figure 1 shows that $CH_3CN$ is similar to HCN, although the tape recorder signal is only clear in the 2016-18 period. There was a large influx of both HCN and $CH_3CN$ at that time due to a very strong El Niño event.

[Figure]

figure-1.png

**Fig. 1.** Time series of MLS CH$_3$CN anomaly. The data are zonal means covering the latitude range 15°S to 15°N; the time mean of those zonal means has been subtracted.

**3  Technical corrections**

- Page 1 Line 1: Maybe replace "one of the volatile organic compounds" with "a volatile organic compound".

- Page 1 line 7: Maybe replace "a pressure broadening" with "pressure broadening" or "the pressure broadening coefficient".

- Page 2 line 33: "lower stratosphere Kopp and Arnold et al. (1978); Schneider et al. (1997)." should be "lower stratosphere (Kopp and Arnold et al., 1978; Schneider et al.,1997)" (In LaTeX this would be a `\citep`, not a `\citet`.)

- Page 2 Lines 51-53: Figures should be called out in numerical order. Here, Figure 2 is mentioned in the running text before Figure 1.

- Page 17 line 216: AMT prefers datasets to be referenced in the same way as papers, with the DOI included. The full reference for the MLS CH$_3$CN data is Santee and Read (2015).

**References**

Mote, P. W., Rosenlof, K. H., McIntyre, M. E., Carr, E. S., Gille, J. C., Holton, J. R., Kinnersley, J. S., Pumphrey, H. C., Russell, J. M., and Waters, J. W.: An atmospheric tape recorder: The imprint of tropical tropopause temperatures on stratospheric water vapor, J. Geophys. Res, 101, 3989–4006, https://doi.org/10.1029/95JD03422, 1996.

Pumphrey, H. C., Boone, C., Walker, K. A., Bernath, P., and Livesey, N. J.: Tropical tape recorder observed in HCN, Geophys. Res. Lett, 35, L05801, https://doi.org/10.1029/2007GL032137, 2008.

Pumphrey, H. C., Glatthor, N., Bernath, P. F., Boone, C. D., Hannigan, J., Ortega, I., Livesey, N. J., and Read, W. G.: MLS measurements of stratospheric hydrogen cyanide during the

2015–16 El Niño event, Atmospheric Chemistry and Physics, 2018, 691–703, https://doi.org/10.5194/acp-18-691-2018, 2018.

Rodgers, C. D.: Inverse Methods for Atmospheric Sounding: Theory and practise, World Scientific, ISBN 981-02-2740-X, 2000.

Santee, M. L. and Read, W.: MLS/Aura Level 2 Methyl Cyanide (CH3CN) Mixing Ratio V004, https://doi.org/10.5067/Aura/MLS/DATA2003, date accessed: (Insert date here), 2015.

---

## Editor Comment (EC1) · Alyn Lambert (Editor) · 1 Nov 2019

Here is the missing Figure 1 from page C4 of the review provided by RC2.

[Figure]

[Figure]

**Fig. 1.** Please see the RC2 review page C4 for context.

---

## Author Comment (AC1) · 12 Dec 2019

Dear Anonymous Reviewer #1,

We deeply appreciate for valuable comments and suggestions. Please find the manuscript with several revisions. Answers to your comments/questions are given by point by point. We hope that current manuscript is significant for the publication in Atmospheric Measurement Techniques.

Sincerely yours, Tamaki Fujinawa.

[Figure]

General comments 1-1

The level of English in the manuscript could use some work, starting with the title. It is readable, but the phrasing is occasionally not quite correct. There are too many instances for me to provide line-by-line corrections.

Answer to general comments 1-1

We appreciate your valuable comment. As you mentioned, we have checked the grammar and phrasing again.

Major comment 1-2

I have some issues with the background discussion on CH3CN. While it is all technically correct that the molecule is associated with biomass burning and pollution and can be used as a tracer for biomass burning events, most biomass burning activity occurs in the troposphere but the measurements reported here are in the stratosphere. Some of the discussion is not entirely relevant to what they are measuring. They report the observation of a seasonal maximum of CH3CN in the upper stratosphere in February supposedly resulting from a peak in biomass burning during the time period from December to March. Age of air in the upper stratosphere is the order of a few years, but they appear to be suggesting they are measuring enhanced CH3CN in the upper stratosphere almost immediately. It should take a few years for young tropospheric air containing enhanced CH3CN to make its way to that atmospheric region. Intense fires can inject biomass burning products directly into the stratosphere (pyro-convection), but I don't think that is what they propose is happening here. CH3CN levels would be anomalously high in that case, and I expect the effect would not extend into the upper stratosphere. It would be confined to the lower stratosphere. The authors may want to rethink their interpretation taking transport times to the upper stratosphere into consideration.

Answer to major comment 1-2

We are grateful for pointing this out and for your valuable comment. In our understanding, we could assume that CH3CN is almost emitted from biomass burning into the atmosphere and there is no major source of CH3CN in the stratosphere. In addition, it is likely that there is seasonality of CH3CN in lower stratosphere from 28 km to 36 km as can be seen in Figure 7, even if the error bars (one sigma standard deviation) were taken into account. Therefore, we thought that the enhancement which can be seen in Figure 7 would be caused by biomass burning event. However, as you mentioned, our description about the enhancement might cause a misunderstanding that biomass burning plume emitted during the time period from December to March was immediately transported to the stratosphere. Therefore we revised some parts of the sentences as follows to make it clear.

Revisions to major comment 1-2

Lines 5-6: "We estimated the systematic and random errors to be ∼5.8 ppt (7.8 %) and 25 ppt (60 %) for a single observation at 15.7hPa, respectively, in the Tropics, where the CH3CN measurements are enhanced." was replaced by

→

Lines 5-6: "We estimated the systematic and random errors to be ∼5.8 ppt (7.8 %) and 25 ppt (60 %) for a single observation at 15.7hPa, respectively, in the Tropics.".

Lines 198-199: ", which is consistent with our understanding that most biomass burning occurs from December–March." was removed.

Lines 198-199: "These seasonal maximum CH3CN levels measured in February are consistent with the yearly high period of BB events from December–March." was removed.

Major comment 1-3

In Figure 7, for altitudes below 40 km, the systematic high bias of results from AOS1 relative to results from AOS2 are quite evident. The authors make the statement that

the differences between the AOSs is due to "sensitivity differences." I do not know what that means. The phrasing would suggest something like signal-to-noise ratio, but I would expect that to yield increased variability and not a systematic offset, unless perhaps the retrieval is being "pulled" more toward the a priori in the optimal estimation analysis because of the reduced signal-to-noise ratio. Is it possible to provide a few extra words to explain what is meant by sensitivity differences? It might be instructive to see the signal from the two different AOSs under similar measurement conditions, to see why one is yielding a larger VMR than the other.

Answer to major comment 1-3

We appreciate your valuable comment. The "sensitivity differences" indicate inherent sensitivity differences between the two AOSs derived from instrumental characterization determined when manufacturing. As you pointed out, only using of the phrase "sensitivity differences" was unclear. Therefore, we added an additional explanation as below.

Revisions to major comment 1-3

Line 135 : "Note that the sensitivity differences indicate inherent sensitivity differences between the two AOSs derived from instrumental characterization determined when manufacturing." was added.

Major comment 1-4

In Figure 9, the top VMR two points from MLS are negative, which contributes to the strong magnification of the discrepancies between the two instruments in this altitude region. Negative VMRs have no physical meaning. You should probably point this out when discussing the discrepancies. This is an item in your favor.

Answer to major comment 1-4

We appreciate you pointing this out. As you mentioned, it shows the exaggeration of the discrepancies between the two observation. We added a sentence as below to

point that out.

Revisions to major comment 1-4

Line 180 : "It should be noted that the discrepancies between the two instruments were exaggerated at upper pressure levels from 2.8\,hPa although negative values of CH3CN VMR derived from MLS have no physical meaning." was added.

Major comment 1-5

How was the a priori state constructed? If MLS results went into generating the a priori used in the optimal estimation analysis, extra care might be required in using comparisons to MLS as part of a validation effort.

Answer to major comment 1-5

We appreciate your valuable comment. We used the results from version 5.2 of the Goddard Earth Observing System Model (GEOS-5.2) as a priori information (e.g., O$_3$ VMR profile, temperature and pressure profile). Therefore, we think it is not required to provide extra care about it. However, we added an explanation about preparing a priori state as you pointed out as below.

Revisions to major comment 1-5

Line 57 : "This version of L2r product was derived from the Level-1b (L1b) version 008 calibrated spectra, which used version 5.2 of the Goddard Earth Observing System Model (GEOS-5.2) as a priori information (e.g., O3 VMR profile, temperature and pressure profile) (Rienecker et al., 2008)." was added.

––––––––––––––––––––––––––––––––

---

## Author Comment (AC2) · 12 Dec 2019

Dear Dr. Hugh C. Pumphrey,

We deeply appreciate for valuable comments and suggestions. Please find the manuscript with several revisions. Answers to your comments/questions are given by point by point. We hope that current manuscript is significant for the publication in Atmospheric Measurement Techniques.

Sincerely yours, Tamaki Fujinawa.

[Figure]

Specific comment 1-1

Page 2 line 25: "Acetonitrile [. . . ] is one of the dominant gases emitted during wildfire events (90%-95%)". This statement is either wrong or confusing as wildfires emit far more CO and CO2 than they do CH3CN. What I suspect that the authors mean is that 90%-95% of CH3CN comes from wildfires. If that is what the authors mean, they should say so explicitly.

Answer to specific comment 1-1

We appreciate your comment. As you mentioned, the sentence is confusing. To avoid misunderstanding, we replace it as below.

Revisions to specific comment 1-1

Line 25: "[. . .] as it is one of the dominant gases emitted during wildfire events (90–95 %)" was replaced by

$\rightarrow$

Line 25: "as 90-95 % of CH3CN comes from wildfires".

Specific comment 1-2

Page 2 line 27:"...ocean uptake and the reaction with hydroxyl radicals (OH)([. . . refs . . . ])." This sentence appears to be missing some words at the end. Maybe it should read "...ocean uptake and the reaction with hydroxyl radicals (OH)([. . . refs . . . ]) being the main loss mechanisms."

Answer to specific comment 1-2

We appreciate your valuable comment. As you mentioned, some words which you recommended is added as below.

Revisions to specific comment 1-2

Line 27 : "[. . .] being the main loss mechanisms." was added.

Specific comment 1-3

Page 5 line 58 "maximum a posteriori solution" should be "maximum a posteriori probability solution". When mentioning the MAP solution it is surely obligatory to cite Rodgers (2000).

Answer to specific comment 1-3

We appreciate you pointing this out. As you mentioned, we add the word 'probability' and cite Rodgers (2000).

Revisions to specific comment 1-3

Line 58 : "maximum a posteriori solution" was replaced by "maximum a posteriori probability solution (Rodgers, 2000)"

Specific comment 1-4

Page 16, Figure 10. I struggle to interpret this figure. This is partly because the individual panels are rather small; I do not know what might be done about this as the arrangement of the figure is useful. The MLS data appear to have a lot of gaps in; the number of gaps increases with altitude. Now, it is clear that some of the gaps are there because the authors have chosen to show MLS data only for times when there is also SMILES data available. I would suggest that for the middle column of data they might want to show MLS data for all days, so that the eye can more easily pick out the patterns in the data. The actual MLS data do not become more sparse with altitude as the figure suggests: I just plotted them up to check. However, they do become negative on average, as can be seen in Figure 9 (left). I would suggest that the authors use a colour scale which spans the range of both the MLS and SMILES data. Clearly, the MLS data are wrong in the sense that the atmosphere can not contain less than none of a constituent. But the time-latitude dependence of the MLS data is actually rather similar to that of the SMILES data — it would be nice if the plot could bring that out. I am pleased to see a sequential colour scale used for the actual quantities and a diverging

one for the SMILES-MLS differences. Less satisfactory is the choice of sequential colour scale; the authors have used the notorious "jet" scale, or something very like it. (See https://hughpumphrey.wordpress.com/2017/06/29/ colours-for-contours/ for some thoughts and some useful links.) They might want to consider whether a scale other than "jet" might be appropriate in this figure.

Answer to specific comment 1-4

We deeply appreciate your valuable comment and link. As you mentioned, we revised Fig. 10 about the following points: (1) in the middle column in Fig. 10, the MLS data for all day, which is including the not-coincident points, were considered, (2) the color scale was expanded to negative values, (3) the sequential color map of 'jet' was changed to 'magma'. We hope these revisions help you to interpret this figure.

Revisions to specific comment 1-4

Please see the revised Figure 10 (Fig. 1 HERE). Revised points are as following. (1) In the middle column in Fig. 10, the MLS data for all day, which is including the not-coincident points, were considered. (2) the color scale was expanded to negative values. (3) the sequential color map of 'jet' was changed to 'magma'.

Specific comment 1-5

Page 16 Lines 194-196: The time-latitude dependence of the MLS data is actually quite clear and easy to see as long as you plot the data up with a colour scale that goes into the negative. Clearly, the MLS data have a negative bias of over 100% in the upper stratosphere, but this does not, of itself, prevent the seasonal behaviour being observed.

Answer to specific comment 1-5

We are grateful for pointing this out. We expanded the color scale of the MLS data to negative values at upper pressure levels in Fig. 10. As you mentioned, results of the MLS also showed the time-latitude dependence clearly in the upper stratosphere

despite of the large negative bias. We think this comment is partly included in the specific comment 2-4. Therefore, please see also the answer and revisions against specific comment 2-4 in detail.

Revisions to specific comment 1-5

This comment seems to be almost same as Specific comment 1-4. Please see the revised Figure 10.

Specific comment 1-6

Page 16 Lines 198-199 and page 17 lines 210-211: It is not at all clear to me why there should be a connection between the maximum observed at 1hPa – 5 hPa in February and the timing of the biomass-burning season. Tropospheric source gases such as CH3CN enter the stratosphere via the tropical tropopause and take over a year to ascend from 100hPa (16km) to 10hPa (32km). This "tape recorder" effect was first observed in water vapour (Mote et al., 1996) and subseqently in HCN (Pumphrey et al., 2008, 2018) among other species. Figure 1 shows that CH3CN is similar to HCN, although the tape recorder signal is only clear in the 2016-18 period. There was a large influx of both HCN and CH3CN at that time due to a very strong El Niño event.

Answer to specific comment 1-6

We deeply appreciate pointing this out and your valuable comment. The other anonymous referee also pointed out the same point. We thought there is seasonality of CH3CN in lower stratosphere from 28 km to 36 km as can be seen in Figure 7, even if the error bars (one sigma standard deviation) were taken into account. Therefore, we thought that the enhancement which can be seen in Figure 7 would be caused by biomass burning event. However, as you mentioned, our description about the enhancement might cause a misunderstanding that biomass burning plume emitted during December to March directly caused the seasonal maximum measured in February.

Revisions to specific comment 1-6

Lines 5-6: "We estimated the systematic and random errors to be ∼5.8 ppt (7.8 %) and 25 ppt (60 %) for a single observation at 15.7hPa, respectively, in the Tropics, where the CH3CN measurements are enhanced." was replaced by

→

Lines 5-6: "We estimated the systematic and random errors to be ∼5.8 ppt (7.8 %) and 25 ppt (60 %) for a single observation at 15.7hPa, respectively, in the Tropics.".

Lines 198-199: ", which is consistent with our understanding that most biomass burning occurs from December–March." was removed.

Lines 198-199: "These seasonal maximum CH3CN levels measured in February are consistent with the yearly high period of BB events from December–March." was removed.

Technical correction 2-1∼2-5

Page 1 Line 1: Maybe replace "one of the volatile organic compounds" with "a volatile organic compound".

Page 1 line 7: Maybe replace "a pressure broadening" with "pressure broadening" or "the pressure broadening coefficient".

Page 2 line 33: "lower stratosphere Kopp and Arnold et al. (1978); Schneider et al. (1997)." should be "lower stratosphere (Kopp and Arnold et al., 1978; Schneider et al.,1997)" (In LATEX this would be a \citep, not a \citet.)

Page 2 Lines 51-53: Figures should be called out in numerical order. Here, Figure 2 is mentioned in the running text before Figure 1.

Page 17 line 216: AMT prefers datasets to be referenced in the same way as papers, with the DOI included. The full reference for the MLS CH3CN data is Santee and Read (2015).

[Figure]

Answer to technical correction 2-1∼2-5

We appreciate your valuable suggestions. As you mentioned, we revised several points as following. Detailed revisions are described in "Revision to technical correction 3-1∼3-5".

Revisions to technical correction 2-1∼2-5

Page 1 Line 1: "[. . .] one of the volatile compounds [. . .] " was replaced by

→

"[. . .] a volatile organic compound [. . .]"

Page 1 line 7: "[. . .] a pressure broadening [. . .]" was replaced by

→

"[. . .] the pressure broadening coefficient [. . .]"

Page 2 line 33: "lower stratosphere Kopp and Arnold et al. (1978); Schneider et al. (1997)." was replaced by

→

"lower stratosphere (Kopp and Arnold et al., 1978; Schneider et al.,1997)"

Page 2 Lines 51-53: "[. . .] as shown in Fig. 2. SMILES employed [. . .] The date of observation made by AOS1 and AOS2 are shown in Fig. 1." was replaced by

→

"[. . .] as shown in Fig. 1. SMILES employed [. . .] The date of observation made by AOS1 and AOS2 are shown in Fig. 2."

[Figure]

**Fig. 1.** Revised figure from Fig. 10 in the manuscript.

---

## Author Response (AR2)

**Second Reply to Anonymous Reviewer #1**

Dear Anonymous Reviewer #1,

We appreciate for giving us valuable comments and suggestions. Please find the manuscript, which has been done for native-checking of the grammar and revised according to your suggestions. There are a lot of revisions w.r.t incorrect grammar, so that we cannot list by point by point. As following, we answer to, and list only about major revisions related with your comments. We hope that the current manuscript become suitable for the publication in Atmospheric Measurement Techniques.

Sincerely yours,

[1]Tamaki Fujinawa and [2]Yasuko Kasai
[1]National Institute for Environmental Studies
[2]National Institute of Information and Communications Technology

**Comments 1-1**

I will make a suggestion for the title:

Validation of acetonitrile (CH3CN) measurements in the stratosphere and lower mesosphere from the SMILES instrument on the International Space Station

**Answer to Comments 1-1**

We appreciate your suggestion. As you suggest, we have changed the title as the following.

**Revisions to Comment 1-1**

'Validation analysis of deriving acetonitrile ($¥chem{CH_{3}CN}$) profiles by observations of SMILES from the International Space Station, in the stratosphere and lower mesosphere' was changed as

'Validation of acetonitrile (CH$_3$CN) measurements in the stratosphere and lower mesosphere from the SMILES instrument on the International Space Station'.

**Comment 1-2**

I will point out a sentence that requires rewriting:

The difference between the two decreases down to less than 10 % at an upper altitudes than 4.8 hPa.

**Answer to Comment 1-2**

As you suggest, we have replaced this sentence as the following.

**Revisions to Comment 1-2**

Lines 140: "The difference between the two decreases down to less than 10 % at an upper altitudes than 4.8 hPa." was replaced by

→

"The relative difference between the two AOSs decreases to less than 10 % at an upper altitude than 4.8 hPa, except at 0.3 hPa, showing a good agreement of the two AOS observations from the middle stratosphere".

**Comment 1-3**

The word "indicating" suggests that the magnitude of the difference and/or the pressure level at which the maximum difference occurs somehow proves that the discrepancy is due to "sensitivity differences." I don't really follow that logic. At the very least, I think you should rephrase ("We believe the discrepancy is due to sensitivity differences in the two AOSs" or "It

is likely that these discrepancies between the two AOSs result from sensitivity differences")
to something less definitive, unless there is proof provided here that I am missing.

You seem to basically attribute the inconsistencies between the AOSs to problems in the
instrument characterization during manufacturing. I don't know if this means you see similar
effects (higher VMR from AOS 1) for all molecules, weak or strong, from band A, or whether
the problem relates to difficulties in properly characterizing instrumental effects in the shape
of the stronger, overlapping HCl spectral feature, which then pollutes the retrieval using the
weak CH3CN spectral feature. I am imagining it is the latter.
This would be beyond the scope of this paper, but as food for thought, would it be possible to
refine the instrument characterization to achieve improved agreement between the two
instruments? I don't know how many moving parts go into that characterization.

**Answer to Comment 1-3**

  We appreciate pointing this out. As you comment, we replaced some words, and added some
descriptions about the inconsistencies between the two AOSs, as below.

**Revisions to major comment 1-3**

Line 136 and following : "Note that the sensitivity differences indicate inherent sensitivity
differences between the two AOSs derived from instrumental characterization determined
when manufacturing." was replaced by
'We believe that the sensitivity differences indicate inherent sensitivity differences between
the two AOSs derived from instrumental characterization determined during manufacturing.'.

The sentences,
'Kasai et al. (2013) also reported the discrepancies between the two different AOSs, albeit
for the analysis of ozone profiles using the SMILES L2 version 2.1.5 product.
As mentioned above, in this analysis, we used the SMILES L2r version 3.0.0 product that
improves the AOS response function.
However, there may still be disagreement between the two AOSs.
The relative difference between the two AOSs decreases to less than 10¥,¥% at an upper
altitude than 4.8¥,hPa, except at 0.3¥,hPa, showing a good agreement of the two AOS
observations from the middle stratosphere.' was added.